# Gene Expression Profiles Induced by a Novel Selective Peroxisome Proliferator-Activated Receptor α Modulator (SPPARMα) Pemafibrate

**DOI:** 10.3390/ijms20225682

**Published:** 2019-11-13

**Authors:** Yusuke Sasaki, Sana Raza-Iqbal, Toshiya Tanaka, Kentaro Murakami, Motonobu Anai, Tsuyoshi Osawa, Yoshihiro Matsumura, Juro Sakai, Tatsuhiko Kodama

**Affiliations:** 1Laboratories for Systems Biology and Medicine (LSBM), Research Center for Advanced Science and Technology (RCAST), The University of Tokyo, Tokyo 153-8904, Japan; y-sasaki@kowa.co.jp (Y.S.); sanaraza501@yahoo.com (S.R.-I.); k-murakm@lsbm.org (K.M.); anai@lsbm.org (M.A.); kodama@lsbm.org (T.K.); 2Tokyo New Drug Research Laboratories, Kowa Company, Ltd., Tokyo 189-0022, Japan; 3Division of Integrative Nutiriomics and Oncology, Research Center for Advanced Science and Technology (RCAST), The University of Tokyo, Tokyo 153-8904, Japan; osawa@lsbm.org; 4Division of Metabolic Medicine, Research Center for Advanced Science and Technology (RCAST), The University of Tokyo, Tokyo 153-8904, Japan; matsumura-y@lsbm.org (Y.M.); jmsakai@lsbm.org (J.S.); 5Tohoku University of Graduate School of Medicine, Division of Molecular Physiology and Metabolism, Sendai 980-8575, Japan

**Keywords:** pemafibrate, SPPARMα, ketogenesis, fatty acid β-oxidation, ASCVD, EndMT

## Abstract

Pemafibrate is the first clinically-available selective peroxisome proliferator-activated receptor α modulator (SPPARMα) that has been shown to effectively improve hypertriglyceridemia and low high-density lipoprotein cholesterol (HDL-C) levels. Global gene expression analysis reveals that the activation of PPARα by pemafibrate induces fatty acid (FA) uptake, binding, and mitochondrial or peroxisomal oxidation as well as ketogenesis in mouse liver. Pemafibrate most profoundly induces *HMGCS2* and *PDK4*, which regulate the rate-limiting step of ketogenesis and glucose oxidation, respectively, compared to other fatty acid metabolic genes in human hepatocytes. This suggests that PPARα plays a crucial role in nutrient flux in the human liver. Additionally, pemafibrate induces clinically favorable genes, such as *ABCA1*, *FGF21*, and *VLDLR*. Furthermore, pemafibrate shows anti-inflammatory effects in vascular endothelial cells. Pemafibrate is predicted to exhibit beneficial effects in patients with atherogenic dyslipidemia and diabetic microvascular complications.

## 1. Introduction

Although low density lipoprotein cholesterol (LDL-C)-lowering therapy by statins has been proven to reduce the events of atherosclerotic cardiovascular disease (ASCVD) [1,2], there still remains a high residual cardiovascular risk from elevated triglycerides (TG) and low HDL cholesterol (HDL-C) levels [3,4,5,6]. Synthetic PPARα ligands and fibrates have been shown to effectively reduce plasma TG levels by 25–50% and increase HDL-C levels by 5–20% [7,8,9,10]. Therefore, theoretically, fibrates are suitable drugs to use as an add-on statin treatment to improve hypertriglyceridemia and atherogenic dyslipidemia. However, there is a lack of adequate evidence to support statin-fibrate combination therapy for the prevention of definitive mortality rate. In addition, the use of fibrates in patients with hepatic and renal insufficiency has been limited due to adverse drug reactions (ADRs) such as plasma transaminase and creatinine elevation, as well as reduced estimated glomerular filtration rates (eGFRs) [11,12,13,14]. Under these circumstances, pemafibrate was developed as a selective peroxisome proliferator-activated receptor α modulator (SPPARMα) that enhances the beneficial effects and reduces the adverse effects of fibrates. To date, 50 papers have been published on this subject and few papers reported the effect of pemafibrate on target gene expression. Through the limited reports, we describe the pemafibrate-regulated genes and potential clinical implications.

## 2. Pemafibrate as a Novel SPPARMα

Pemafibrate (K-877, Parmodia^®^) was developed as a novel SPPARMα that enhances PPARα activity and selectivity by introducing a 2-aminobenzoxazolic ring and phenoxyalkyl chain into fibric acid (Figure 1a) [15,16,17]. These side-chains confer a Y-shape structure and fill the entire ligand-binding pocket of PPARα [18] (Figure 1b), thereby allosterically changing the PPARα conformation to enhance complex formation with coactivators such as peroxisome proliferative activated receptor gamma coactivator 1α (*PGC1α*) and exhibiting full agonistic activity. Actually, pemafibrate has greater PPARα activation potency than fenofibrate, along with a lower EC_50_ value (1.5 nM) and a higher degree of subtype selectivity (>2000-fold) (Figure 1c) [19]. In preclinical studies, pemafibrate exhibited a greater TG-lowering effect than fenofibrate in normolipidemic and hypertriglyceridemic rodent models [15,20,21]. In addition, in human apoA-I transgenic mice, pemafibrate treatment resulted in a greater increase in levels of plasma h-apoAI, a major component of HDL, than occurred with fenofibrate treatment [15,22]. Furthermore, pemafibrate has been shown to reduce atherosclerotic lesion areas in *Ldlr*-null mice [17] and western diet-fed APOE2 KI mice [22]. Although fibrates have been specifically shown to induce peroxisome proliferation and related hepatomegaly and hepatocellular carcinoma in rodents [23,24,25], pemafibrate causes less weight gain of the liver than fenofibrate [15]. Under the fed condition, the liver accumulated the highest concentration of pemafibrate and reached 105 nM after four weeks of treatment with a 0.0006% (*w*/*w*) pemafibrate-containing diet, which is an equivalent or higher dose than needed to demonstrate pharmacological action [22,26,27]. As indicated in Figure 1c, pemafibrate was unable to activate PPARγ or PPARδ at this concentration. In addition, the therapeutic dose of pemafibrate is 0.2–0.4 mg/day, which is equivalent to the dose of 0.004–0.008 mg/kg/day (based on a 50 kg human); therefore, it is unlikely that pemafibrate shows the other PPARs subtype-mediated pharmacological effect in clinical use.

Pemafibrate was approved in Japan 2017 for the treatment of dyslipidemia [28,29,30,31,32,33,34,35,36,37,38]. A phase II study showed that 0.05–0.4 mg/day pemafibrate significantly reduced plasma TG levels (−30.9% to −42.7%) and increased HDL-C levels (11.9% to 21.0%) [29]. Although the difference was not statistically significant, the improvement of these parameters was more significant with pemafibrate than fenofibrate. The incidence of adverse events (AEs) in the pemafibrate treatment group was comparable to those in the placebo and 100 mg/day fenofibrate groups. However, the incidence of ADRs in the pemafibrate treatment group was lower than those in the placebo and 100 mg/day fenofibrate groups [29,31]. In addition, when compared to placebo and fenofibrate treatment, pemafibrate significantly increased the level of plasma FGF21, which is an endocrine factor regulating glucose uptake, metabolism, and energy expenditure [39]. Therefore, pemafibrate could replace fibrates as the first clinically-available SPPARMα to improve atherogenic dyslipidemia and prevent macro- and microvascular risks.

## 3. Pemafibrate Regulates the Availability of FA and Glucose Oxidation

Species differences have been well documented for PPARα-regulated genes, such as those involved in peroxisome biogenesis and peroxisomal FA β-oxidation [40,41,42]. In addition, whether PPARα mediates gene expression regulation by pemafibrate and whether human exposure to pemafibrate regulates the same target genes as those found in mice are still a matter of debate. To predict the mode of action and untoward effects of pemafibrate in humans, we carried out microarray analyses and compared the data of pemafibrate-treated primary human hepatocytes and mouse livers [19].

Global gene expression profiling clearly demonstrated that pemafibrate regulates the entire FA catabolism in mouse liver. Pemafibrate significantly induces *Vldlr*, TG hydrolysis (*Lpl*), FA cellular uptake (*Cd36/Fat*, *Slc27a1*, and *Slc27a4*), FA binding (*Fabp2* and *Fabp4*), FA activation (*Acsl1, Acsl3, Acsl5*, and *Acot1*), FA ω-oxidation (*Cyp4a14, Cyp4a31*, and *Aldh3a2*), and peroxisomal (*Abcd2, Abcbd3, Ech1, Decr2, Acox1, Ehhadh, Hsd17b4, Acaa1, Crat, Acot3, Acot4*, and *Acot8*) and mitochondrial (*Cpt1, Cpt2, Slc25a20, Acadvl, Acadl, Acads, Acadm, Acad11, Ehhadh, Hadha, Hadhb*, and *Decr1*) FA β-oxidation, and ketogenesis (*Acat1, Hmgcs2*, and *Hmgcl*). In addition, pemafibrate induces peroxisome biogenesis genes (*Pex1, Pex3, Pex11a, Pex14*, and *Pex19*). The upregulation of these genes was not observed in the pemafibrate-treated *Ppara*-null mouse liver [19]. In accordance with our results, Takei et al. also reported that the effect of pemafibrate was abolished in *Ppara*-null mice [21]. Thus, these observations indicate that PPARα is crucial for the regulation of FA catabolic genes in mouse liver following pemafibrate treatment.

Similarly, pemafibrate induced *VLDLR, FABP1*, and mitochondrial FA β-oxidation gene (*ACSL1*, *ACSL5*, *CPT1A*, *CPT2*, *SLC25A20*, *ACADVL*, *HADHA*, *HADHB*, and *ACAA2*) expression in human hepatocytes, as seen in the livers of pemafibrate-treated mice. However, the induction of these genes was much lower in the human hepatocytes (Figure 2). Additionally, pemafibrate did not induce almost all FA ω-oxidation, peroxisomal FA β-oxidation, and peroxisome biogenesis genes expressions. The first step of FA ω-oxidation is ω-hydroxylation, which is catalyzed by the CYP4A family. Generated products are further metabolized to dicarboxylic acid by cytosolic aldehyde dehydrogenase, which is encoded by *ALDH3A2*, and they are efficiently metabolized by peroxisomal FA β-oxidation [43,44]. Numerous reports clearly indicated that the CYP4A family of enzymes are regulated by PPARα in rodent livers and are shown to parallel the induction of peroxisomal fatty acid β-oxidation enzymes and peroxisome proliferation [45]. In contrast, respect to the induction of CYP4A subtype is controversial in humans. Some studies showed that fibrates induce CYP4A11 mRNA expression in primary human hepatocytes and PPARα overexpressed HepG2 cells [46,47]. However, 100 μM of fenofibric acid, a concentration which is equal with our previous study, has been reported to fail induction of CYP4A11 expression in HepG2 cells [41]. Although it is difficult to declare the possibility to induce FA ω-oxidation enzyme in humans at present, peroxisome proliferation and related liver toxicities would not occur following a clinical dose of pemafibrate treatment.

Interestingly, pemafibrate most profoundly induced *PDK4* and *HMGCS2* gene expression in the primary human hepatocytes. Robust induction of *PDK4* indicated inactivation of pyruvate dehydrogenase (PDH) and glucose oxidation [48,49,50]. In contrast, HMGCS2 expression has been reported to control not only ketogenesis but also mitochondrial fatty acid oxidation in HepG2 cells [51]. In addition, this report also showed that the expression of FGF21 (another target of pemafibrate) is upregulated by HMGCS2 activity or acetoacetate, which is the oxidized form of the ketone bodies. Furthermore, the ketone body, β-hydroxybutyrate, as an inhibitor of class I histone deacetylases (HDAC), and β-hydroxybutyrate-integrated histone H3 lysine 9 (H3K9bhb) are associated with the upregulation of genes involved in the starvation-responsive pathways, including the PPAR signaling pathway [52]. Thus, PPARα activation by pemafibrate cooperatively regulates nutrient availability through the induction of the key target genes, namely PDK4 and HMGCS2, which suppress the availability of carbohydrate oxidation and enhance acyl-CoA flux. This thereby facilitates mitochondrial long-chain fatty acid β-oxidation and ketogenesis in human hepatocytes. As a result, pemafibrate reduces the availability of acetyl-CoA for de novo lipogenesis and VLDL secretion.

## 4. Pharmacologically Favorable Target Genes of Pemafibrate as a SPPARMα

As shown in Figure 3, compared to fenofibrate, pemafibrate effectively induces the expression of pharmacologically favorable genes, such as very-low-density lipoprotein receptor (*VLDLR*), ATP binding cassette subfamily A member 1 *(ABCA1*), and fibroblast growth factor 21 (*FGF21*), by maximizing PPARα activation [19]. VLDLR is a member of the LDL-receptor family and is expressed in many tissues, including skeletal muscles, heart, and adipose tissues, whereas its expression is very low in the liver, under normal conditions [53,54]. VLDLR binds TG-rich lipoproteins such as chylomicron and VLDL and mediates the uptake of TG-rich lipoproteins by peripheral tissues through LPL-dependent lipolysis or receptor-mediated endocytosis. Importantly, Gao et al. [55] reported that fenofibrate induces liver *Vldlr* expression in a PPARα-dependent manner and that the TG-lowering effect of fenofibrate was abolished in *Vldlr*-null mice. In addition, although LPL is typically not expressed in the adult liver [56], pemafibrate PPARα dependently induced the expression of *Lpl* in the mouse liver. Thus, pemafibrate enhances TG-rich lipoprotein hydrolysis and uptake by coordinated regulation of *Vldlr*, *Lpl*, and *Cd36* expression. ABCA1, a member of the superfamily of ATP-binding cassette (ABC) transporters, regulates the formation and function of HDL by facilitating the efflux of cholesterol and phosphatidylcholine to lipid-poor apoAI [57,58]. In fact, pemafibrate significantly induced ABCA1 and ABCG1 in human primary macrophages and enhanced HDL stimulated cholesterol efflux [22]. ABCA1 not only plays an important role in the initial step of reverse cholesterol transport (RCT) but is also involved in the anti-inflammatory action to suppress the expression of pro-inflammatory factors [59,60]. Therefore, pemafibrate-mediated increased ABCA1 expression could contribute to HDL-C elevation as well as anti-inflammatory and anti-atherosclerotic activities. FGF21 is a member of the fibroblast growth factor family [39,61], and its administration has been shown to reduce fasting plasma glucose, TG, insulin, and glucagon levels in diabetic rhesus monkeys [62]. FGF21 is a direct target of PPARα [63,64], and pemafibrate increases fasting and postprandial FGF21 levels along with improving dyslipidemia in humans [65]. Interestingly, CREBH [66] and HMGCS2 [51], the liver target genes of pemafibrate, have been reported to regulate FGF21 gene expression. Moreover, similar upregulation of *Abca1*, *Crebh*, and *Fgf21* was observed in pemafibrate-treated *Ldlr* knockout mice liver [26]. Thus, pemafibrate enhances the combination of PPARα, CREBH, and HMGCS2 for the regulation of FGF21 expression.

Beyond regulation of nutrient oxidation, pemafibrate induces mannose-binding lectin 2 (*MBL2*) and glutamyl aminopeptidase (*ENPEP*) only in human hepatocytes (Figure 4). MBL is a soluble pattern recognition molecule involved in the humoral innate immune system [67,68]. In consecutive non-diabetic men, the serum MBL concentration was reduced in obese individuals accompanied by low insulin sensitivity and increased levels of inflammatory markers [69]. ENPEP encodes aminopeptidase A (APA), a member of the M1 endopeptidase family, involved in the catabolic pathway of the renin-angiotensin-aldosterone system that converts angiotensin II to angiotensin III [70,71,72]. In an animal study, the loss of function of *ENPEP* led to hypertension, and recombinant APA reduced the systolic blood pressure (SBP) [73]. Moreover, a rare nonsense variant in ENPEP is reported to be associated with increased SBP [74]. Therefore, these additional pemafibrate targets are likely to reduce cardiovascular disease risks.

Dysfunction and injury of vascular endothelial cells play a critical role in the pathogenesis of ASCVD and chronic kidney disease (CKD) [75,76,77]. ASCVD and CKD share common risk factors including hypertension, hyperglycemia, obesity, and dyslipidemia and are associated with endothelial activation and dysfunction. In particular, high glucose-induced reactive oxygen species (ROS) have been shown to be involved in vascular dysfunction via a diacylglycerol (DAG)-protein kinase C (PKC)-dependent activation of nicotinamide adenine dinucleotide phosphate NAD(P)H oxidase pathway. Pemafibrate has been reported to reduce *Fn1*, *Tgfb1*, *Nox4*, and *Ncf1* expression, and reduce DAG level, PKC activity, and oxidative stress marker (urinary 8-OHdG excretion) level in kidneys of diabetic *db*/*db* mice [78]. Pemafibrate also reduces serum starvation induced monocyte chemoattractant protein-1(MCP-1), regulated on activation, normal T cell expressed and secreted (RANTES), interleukin 6 (IL6), and interferon gamma (IFNγ) expression and secretion in human coronary endothelial cells (HCECs) [79]. Besides its role in inflammation and ROS production, we found that pemafibrate suppresses high glucose-induced endothelial-mesenchymal transition (EndMT) in human umbilical vein endothelial cells (HUVECs). EndMT has emerged as an important process in the pathobiology of valve calcification, myocardial fibrosis, macrovascular complications, and microvascular complications such as diabetic nephropathy and retinopathy [80,81,82]. Experimental evidence demonstrated that TGFβ and Wnt/β-catenin signaling play a role in EndMT and may further contribute to tissue fibrosis [83,84,85]. Interestingly, pemafibrate reduces high glucose-induced *TGFB2*, *COL1A2*, *CX3CL1*, *VCAM1* and *DKK1* expression in HUVECs (Tanaka et al. personal communication). Likewise, fenofibrate has been reported to inhibit TGFβ-induced endothelin-1 (ET-1) expression in human microvascular endothelial cells [86]. ET-1 is a major vasoactive peptide that has been implicated in organ fibrosis through stimulation of EndMT [87,88]. In addition, fenofibrate has been reported to reduce progression of albuminuria and improve diabetic retinopathy [89,90,91]. Therefore, pemafibrate would be expected to prevent endothelial activation and dysfunction, thereby revealing protective effects against diabetic retinopathy, nephropathy, neuropathy, and ASCVD.

## 5. Possible Mechanism for the Gene Expression Regulation Induced by Pemafibrate?

Finally, we will discuss a potential mechanism for transcriptional regulation of hepatic target genes via PPARα activation by pemafibrate. As described in the text, PPARα activation by pemafibrate not only activates transcription of hepatic lipid metabolism genes, but also represses transcription of pro-inflammatory and EndMT-related genes. From the numerous observations, several models have been proposed for gene transcriptional regulation induced by PPARα [92,93,94]. In particular, PPARα functions as obligate heterodimers with retinoid X receptor (RXR). Ligand activated PPARα-RXR heterodimer mainly binds to DR1 elements termed PPAR response elements (PPREs) and recruits numerous coactivators, including CBP/p300 and SRC/p160 family, which contain histone acetyl transferase (HAT) activity, mediators, and the transcriptional preinitiation complex (PIC) [95,96,97,98]. This mechanism explains the main PPARα-dependent transactivation because DNA binding domain (DBD) mutant of PPARα (PPARα_DISS_), which maintains heterodimerization and coactivator interaction ability, lost PPRE binding and transactivation of PPRE-driven reporter genes [99]. On the other hand, transcriptional repression by PPARα is mainly mediated through protein-protein interactions. Ligand-activated PPARα has been reported to directly interact with pro-inflammatory transcription factor p65 and c-Jun, thereby suppressing their target genes such as IL6 and TNFα [100,101,102]. Interestingly, transcriptional repression ability is retained in PPARα_DISS_, indicating PPARα-dependent transrepression of the pro-inflammatory signaling pathway is PPRE-independent [99]. In addition, ligand-activated PPARα binds to coactivator of GRIP1/TIF2, thereby interfering with the C/EBPβ-induced fibrinogen-β gene transcription [103]. Furthermore, several nuclear receptors such as HNF4s, COUP-TFs, and RXR homodimer bind DR1 PPREs and may modulate PPARα-regulated gene expression [104,105,106,107]. Therefore, pemafibrate-induced gene expression appears as a combination of these multiple mechanisms.

## 6. Conclusions

PPARα regulates many hepatic metabolic genes along with lipid and glucose metabolism during prolonged starvation at the transcription levels and produces ketone bodies to provide metabolic fuel for the extrahepatic tissues. Despite accumulating evidence of the residual cardiovascular risks resulting from elevated TGs and lower HDL-C levels, low potent synthetic PPARα agonists (fibrates) have not shown enough evidence to reduce the definitive mortality rate when combined with statin treatment, despite an improvement in dyslipidemia. To overcome this issue, pemafibrate, a more potent and subtype-selective SPPARMα, was developed. By maximizing PPARα activation, pemafibrate effectively enhances TG hydrolysis, FA uptake, FA β-oxidation, and ketogenesis and thereby stimulates plasma TG hydrolysis and reduces VLDL secretion. In addition, pemafibrate enhances ABCA1-mediated HDL neogenesis and prevents the transfer of HDL-cholesteryl esters into TG-rich lipoproteins through the TG-lowering effect of pemafibrate. Through these mechanisms, pemafibrate effectively improves hypertriglyceridemia and low HDL-C levels. Importantly, PPARα activation by pemafibrate induces not only the generation of FAs via TG hydrolysis but also the generation of ketone bodies via FA β-oxidation and ketogenesis. In turn, the FAs could further activate PPARα, and the ketone bodies could promote the transcriptional activity of PPARα. Therefore, pemafibrate is expected to exert strong pharmacological effects and novel therapeutic action through a positive feedback loop and cooperative target gene regulation (Figure 5). In fact, pemafibrate induces clinically favorable key target genes (VLDLR, FGF21, ABCA1, MBL2, and ENPEP) and thereby has the therapeutic potential to address the residual cardiovascular risk. In addition, pemafibrate would expect to show vascular endothelial cell protective effects and prevent diabetic microvascular complications. Currently, a major outcome study, PROMINENT (Pemafibrate to Reduce cardiovascular OutcoMes by reducing triglycerides IN diabetic patiENTs), is underway to investigate whether pemafibrate reduces cardiovascular events in type 2 diabetic patients with atherogenic dyslipidemia [108]. This study will evaluate the role of pemafibrate in the management of residual cardiovascular risk as an add-on therapy to statins.

## Figures and Tables

**Figure 1 ijms-20-05682-f001:**
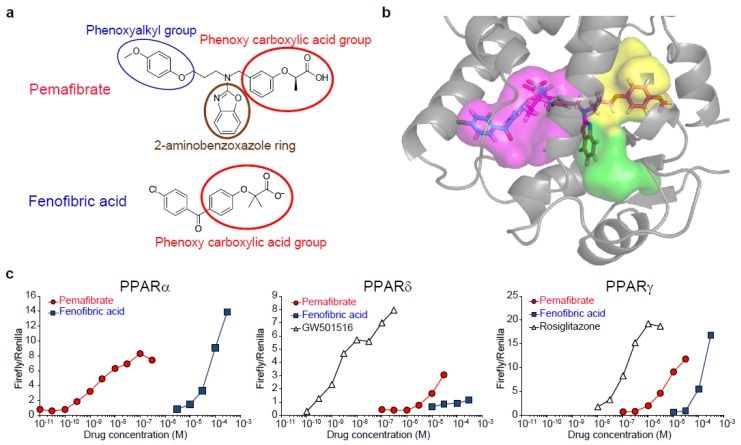
Structure and PPARα selectivity of pemafibrate. (**a**) Structure of pemafibrate and fenofibrate. (**b**) Binding mode of the ligand with human PPARα. Pemafibrate in magenta and fenofibrate in blue. The binding pocket is divided into three pharmacophore regions according to the interactions with the ligands. While fenofibric acid occupies the magenta cavity, 2-aminobenozoxazole ring and phenoxyalkyl group of Y-shaped pemafibrate occupies the green cavity and yellow cavity, respectively. Therefore, pemafibrate fills all the areas of the ligand-binding pocket. Reprinted from Yamamoto Y, et al. with permission from Elsevier [18]. (**c**) Transactivation profile of pemafibrate. Transactivation curves for human PPARα, PPARδ, and PPARγ are shown. Reproduced Raza-Iqbal S., et al. with permission from authors [19].

**Figure 2 ijms-20-05682-f002:**
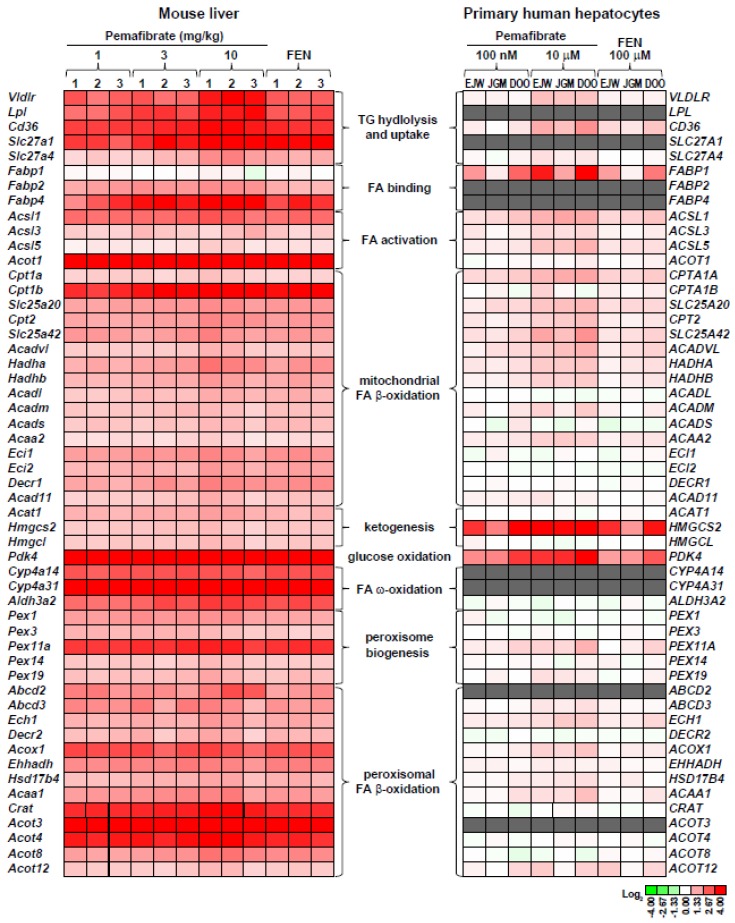
Effect of pemafibrate on fatty acid metabolism-related gene expression. Heat map illustrating the genes regulated by pemafibrate treatment in mouse liver and primary hepatocytes. Gray boxes represent the absence call or no probe of the genes from microarray data.

**Figure 3 ijms-20-05682-f003:**
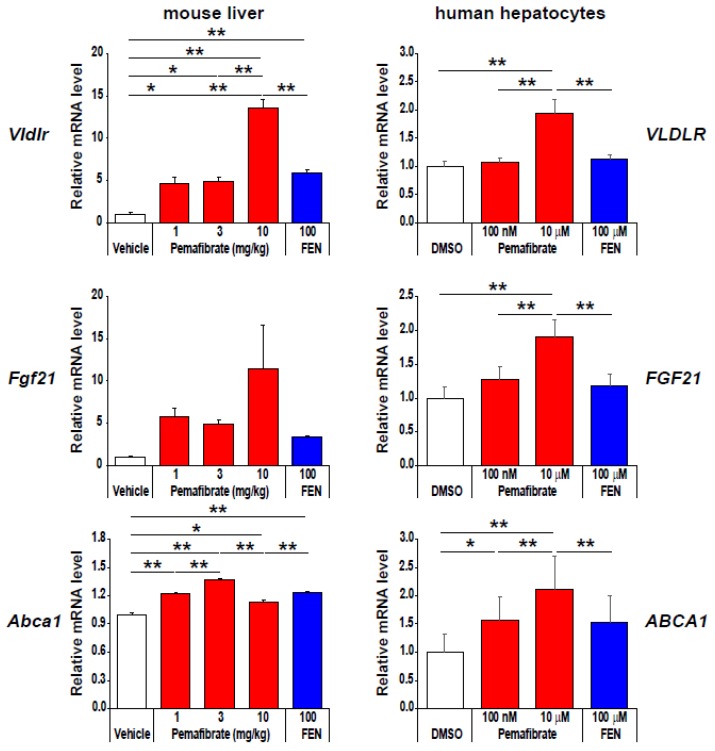
Pemafibrate effectively induces VLDLR, FGF21, and ABCA1 mRNA expression in primary human hepatocytes. Data represent ± s.e.m. * *P* < 0.05; ** *P* < 0.01. Reproduced Raza-Iqbal S., et al. with permission from authors [19].

**Figure 4 ijms-20-05682-f004:**
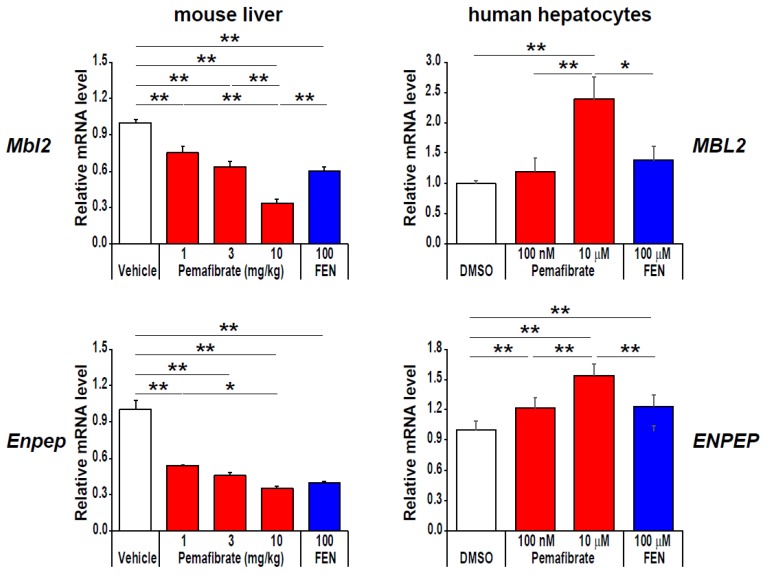
Pemafibrate effectively induces MBL2 and ENPEP mRNA expression in primary human hepatocytes. Data represent ± s.e.m. * *P* < 0.05; ** *P* < 0.01. Reproduced Raza-Iqbal S., et al. with permission from authors [19].

**Figure 5 ijms-20-05682-f005:**
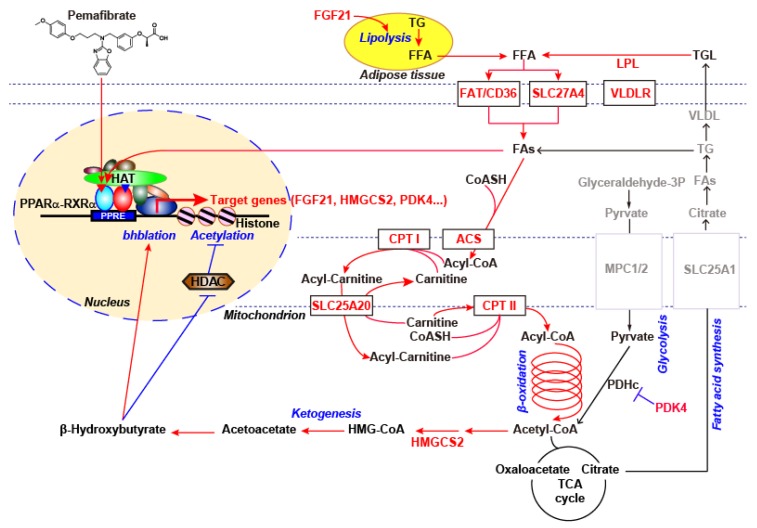
Overviewing pemafibrate regulated fatty acid metabolism genes in human hepatocytes. Red font and arrows indicate the upregulated genes and pathways in the expression microarray of pemafibrate-treated human hepatocytes, respectively, which are based on our microarray data and the published literature.

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
