# Peer review of "Gene Expression Profiles Induced by a Novel Selective Peroxisome Proliferator-Activated Receptor α Modulator (SPPARMα) Pemafibrate"

_ijms, 2019, doi:10.3390/ijms20225682_

Round 1
Reviewer 1 Report
Comment to the Authors:
This manuscript cannot be define as “review” since it does contain only data already published by the same authors in 2015 (J Atheroscler Thromb, 2015; 22:745-7762). For this reason and for the lack of originality I reject the manuscript.
Author Response
This manuscript cannot be define as “review” since it does contain only data already published by the same authors in 2015 (J Atheroscler Thromb, 2015; 22:745-7762). For this reason and for the lack of originality I reject the manuscript.
We thank the reviewer for the valuable comment. As pointed out by the reviewer’s comment, it is true that data present in this manuscript is basically reproduced from our previous J Atheroscler Thromb paper. To date, 50 papers have been reported about pemafibrate, however, the paper which performed comprehensive gene expression analysis of pemafibrate is only our paper. Thus, we wrote this review article about the effect of pemafibrate on liver gene expression based on our previous report. Among these 50 papers, there were the 2 articles which reported influence of pemafibrate on some liver target genes expression, we have sited these 2 articles in the text. In addition, to reveal as the review article, we have added the description about the effect of pemafibrate on endothelial cell gene expression and possible mechanism for the gene expression regulation induced by pemafibrate in the revised manuscript. Moreover, we have added the description for the limitations of this review in the revised text. Because pemafibrate is the novel SPPARMα which was just approved recently, please understand that there is only a limited number of articles performed the gene expression analysis.
Reviewer 2 Report
This paper indicated that SPPARMα enhances PPARα, if SPPARMα also enhances PPAR gamma? Please discuss. Clinical implications need to be further discussed. Please provide limitations of this review. How PARα regulate hepatic metabolic genes? Need to be further address.Author Response
This paper indicated that SPPARMα enhances PPARα, if SPPARMα also enhances PPAR gamma? Please discuss.
Under the fed condition, liver accumulates highest concentration of pemafibrate and reached 105 nM (unpublished data) after 4 weeks treatment with 0.0006% pemafibrate containing diet (equivalent to 0.1 mg/kg/day), which is equal or higher dose to show pharmacological action (Hennuyer, N. et al. Atherosclerosis 2016, 249, 200., Takei, K. et al. J. Pharmacol. Sci. 2017, 133, 214., Sairyo, M. et al. J. Atheroscler. Thromb. 2018, 25, 142.). As indicated in Figure 1, pemafibrate was unable to activate PPARγ or PPARδ at this concentration. In addition, although expression of PPARγ is much lower than PPARα in mice liver, 3 mg/kg/day of pemafibrate-induced hepatic gene regulation was totally abolished in Ppara-null mice. Furthermore, the usual therapeutic dose of pemafibrate is 0.2-0.4 mg/day, which equivalent to dose of 0.004-0.008 mg/kg/day (based on a 50 kg human). Therefore, it is unlikely that pemafibrate shows the PPARγ-mediated pharmacological action at the clinical dose. We have added description about possibility to activate other PPAR subtype in the revised text.
Clinical implications need to be further discussed.
According to reviewers’ comment, to further discuss clinical implications we have added the description about the effect of pemafibrate on endothelial cell gene expression in the revised manuscript. Although unpublished data, we found that pemafibrate inhibits high glucose-induced endothelial-mesenchymal transition (EndMT). EndMT has emerged as an important process in the vascular diseases such as diabetic microvascular complications. Therefore, we expecting the pemafibrate would prevent not only ASCVD, but also diabetic microvascular complications in clinical.
Please provide limitations of this review.
As pointed out by reviewer, presented data on this review is mainly based on our previously reported results and a few available manuscripts which performed gene expression analysis. To date, 50 papers have been reported for pemafibrate, however, the paper which performed comprehensive gene expression analysis of pemafibrate is only our paper. Among these 50 papers, there were 2 articles which reported influence of pemafibrate on some liver target genes expression, we have added description of these 2 articles in the revised text. In addition, we have added the description about effect of pemafibrate on vascular endothelial cells and limitations of this review in the revised text.
How PARα regulate hepatic metabolic genes? Need to be further address.
PPARα activation by ligand not only activate transcription of lipid metabolism gene, but also repress transcription of pro-inflammatory gene. Therefore, several models have been proposed for gene transcriptional regulation induced by PPARα. We have added the description about potential mechanism of transcriptional regulation induced by pemafibrate in the revised text.
Reviewer 3 Report
The review written by Saski et al. shows important profiling of gene expression in response to novel pemafibrate. The review is well written and very informative and organized.
1- The main problem I find here is the description of the interaction between pemafibrate and its target. I strongly recommend adding the structure of PPAR alpha and its interaction with the drug if published. This way the reader can understand clearly how the 2-aminobenzoxazolic ring contributes to the interaction.
2- One more point that needs further explanation more then this short one is why pemafibrate didn't induce almost all fatty acids omega oxidation.
Author Response
1- The main problem I find here is the description of the interaction between pemafibrate and its target. I strongly recommend adding the structure of PPAR alpha and its interaction with the drug if published. This way the reader can understand clearly how the 2-aminobenzoxazolic ring contributes to the interaction.
Thank you very much for your valuable comments. According to the reviewer’s comment, we have added the published structure of pemafibrate and fenofibrate bounded PPARα (Yamamoto, Y. et al. Biochem. Biophys. Res. Commun, 2018, 499, 239.) in revised Figure 1b.
2- One more point that needs further explanation more then this short one is why pemafibrate didn't induce almost all fatty acids omega oxidation.
First step of fatty acid ω-oxidation is ω-hydroxylation, which is catalyzed by microsomal cytochrome P-450 (CYP4) enzymes. Generated products were further metabolized to dicarboxylic acid by cytosolic aldehyde dehydrogenase which is encoded by ALDH3A2, and they are efficiently metabolized by peroxisomal β-oxidation. In fact, several studies indicated that accumulation of non-oxidized acyl-CoA esters by the dysfunction of mitochondrial β-oxidation undergo ω-oxidation, resulting in the production of dicarboxylic acids (Adeva-Andany, MM. et al. Mitochondrion. 2019, 46, 73.), and peroxisomes are essential for the degradation of dicarboxylic acids (Ferdinandusse, S. et al. J. Lipid Res. 2004, 45, 1104.).
Numerous reports clearly indicated that CYP4A family of enzymes are regulated by PPARα in rodent liver, and shown to parallel the induction of peroxisomal fatty acid oxidation enzymes and peroxisome proliferation (Yeldandi, A.V. et al. Mutant Res, 2000, 448, 159.). Importantly, induction of CYP4A1 by peroxisome proliferator methylclofenapate (MCP) has been reported to precede the induction of peroxisomal acyl-CoA oxidase, suggesting that the induction of peroxisomal and microsomal genes are regulated in a coordinate manner (Bell, DR. et al. Biochem. J. 1991, 275, 247.). Furthermore, long-chain dicarboxylic acids formed via the CYP4A1 pathway has been shown to induce peroxisomal β-oxidation and L-FABP (Kaikaus, R.M. et al. J. Biol. Chem. 1993, 268, 9593.). In contrast, respect to the induction of CYP4A subtype is controversial in humans. Some studies showed that fibrates induce CYP4A11 mRNA expression in primary human hepatocytes and PPARα overexpressed HepG2 cells (Raucy, J.L.et al. Toxicol. Sci. 2004, 79, 233. Savas, U. et al. Arch. Biochem. Biophys. 2003, 409, 212.). However, 100 μM of fenofibric acid, which concentration is equal with our study, has been reported to fail induction of CYP4A11 expression in HepG2 cells (Lawrence, J.W. et al. J. Biol. Chem. 2001, 276, 31521.). In addition, Kawashima H. et al. could not identify PPRE in human CYP4A11 genes as seen in rodent CYP4A family (Kawashima, H. et al. Arch. Biochem. Biophys. 2000, 378, 333.).
In our previous studies, we could not observe induction of CYP4As nor ALDH3A2 in primary human hepatocytes. However, we could not declare the possibility that pemafibrate would induce these genes expression in special conditions. From these observations, further studies will be necessary to understand precise physiological importance and species difference between humans and rodents for fatty acid ω-oxidation pathway.
Reviewer 4 Report
Sasaki and colleagues reviewed the effects and gene expression profiles in response to the selective peroxisome proliferator-activated receptor alpha modulator pemafibrate in mouse livers and human hepatocytes. They also put the data in the context of existing clinical studies.
The review is very well written, the topic is novel, and of potential broad interest.
I only want to thank the authors for this well-done work.
Author Response
We thank the reviewer for the very glad comment. We have included new information in the text. Thus, the revised manuscript contains a number of changes, which we believe strengthen the conclusions drawn in the original manuscript.
Round 2
Reviewer 1 Report
Even though the Authors have added two paragraphs I still believe that this manuscript cannot be considered under the “Review” type.
My suggestion is to rethink to the organization of this manuscript by adding some more general studies that have been published with this and other related molecules.
For these reasons I reject again this manuscript.